# Clear tracks or missed connections? A qualitative study exploring how interest-holder perceptions of purpose shape the implementation and experience of the six-month review for stroke survivors

Rich Holmes[1,2]*, Suzanne Ackerley[2,3] Dawn Goodwin[2], Louise A. Connell[2,3]

1 St Richard's Hospital, University Hospitals Sussex NHS Foundation Trust, Chichester, Sussex, United Kingdom, 2 Faculty of Health and Medicine, Lancaster University, Lancaster, Lancashire, United Kingdom, 3 Rakehead Rehabilitation Centre, East Lancashire Hospitals NHS Trust, Burnley, Lancashire, United Kingdom

* richard.holmes8@nhs.net

## Abstract

### Introduction

The six-month review (6MR) for stroke survivors is recommended in clinical guidelines. However, the purpose of the review lacks clarity and has been implemented in variable ways. This study aims to better understand the purpose of the 6MR by comparing and contrasting the perspectives of different interest-holders, and to identify what impact this might have on the functioning of services.

### Method

This paper reports a qualitative analysis drawn from a multiple case study project. Participants were recruited from three interest-holder groups: *Service Providers* (staff members)*, Service Influencers* (managers, commissioners and regional leaders)*,* and *Service Users* (stroke survivors and their carers). Data were collected from semi-structured interviews, clinical observations, and documentary evidence. Interest-holder groups were analysed separately using reflexive thematic analysis. Themes were then compared across interest-holder groups.

### Results

Thirty-six participants were recruited across three interest-holder groups: Service Providers (n = 8), Service Influencers (n = 6), and Service Users (n = 22). Seven themes were identified: two each for Service Providers and Service Influencers, and three for Service Users. Service providers emphasised a desire to deliver person-centred care but were often constrained by systemic pressures. Service

**Editor:** Sebastian Suarez Fuller, University of Oxford Nuffield Department of Clinical Medicine: University of Oxford Nuffield Department of Medicine, UNITED KINGDOM OF GREAT BRITAIN AND NORTHERN IRELAND

**Data availability statement:** All relevant data are within the manuscript and its Supporting information files.

**Funding:** RH is funded via a Clinical Doctoral Research Fellowship from the National Institute for Health and Care Research (Award id: NIHR302163) https://www.nihr.ac.uk/ The views expressed in this publication are those of the authors and not necessarily those of the NIHR, NHS or the UK Department of Health and Social Care. The funders had no role in study design, analysis and interpretation of this study.

**Competing interests:** The authors have declared that no competing interests exist.

influencers saw the review primarily as a mechanism for population-level data collection and service planning, while also acting as a safety net to capture unmet or evolving needs. Service users typically viewed the review as a routine check-up rather than a pivotal moment in the pathway. For some, it marked the end of formal support while others valued the sense of reassurance and validation it provided.

## Conclusion

Interest-holders have differing views on the purpose of the 6MR, with tensions emerging between system-level priorities and person-centred care. Greater clarity on the function of the 6MR may help reduce unwarranted variation in its implementation and ensure it delivers meaningful value to all involved.

## Introduction

Internationally, the lack of a systematic approach to the long-term follow-up of stroke survivors is acknowledged [1]. 'Life after Stroke' care is noted to be an under-researched area with relatively little mention in multiple national guidelines [2]. For several years, UK guidelines have recommended that stroke survivors should have a structured, holistic review of their needs at six months [3–5] termed the six-month review (6MR). This recommendation was later adopted by the European Stroke Action Plan [2]. However, specific guidance on the structure and content of this review is lacking.

A recent survey of practice demonstrated wide variation in how 6MR services have been implemented within England [6]. Identified barriers to the implementation of clinical guidelines in stroke rehabilitation include a lack of specificity and clarity in recommendations, as well as a lack of clinician awareness regarding the theoretical underpinnings of recommendations and how to modify them to local context [7]. These factors may be ameliorated through a better articulation of purpose and the formulation of programme theory to ascertain the key elements of the intervention and the mechanisms through which it is expected to achieve outcomes [8].

The most recent UK guidelines [5] indicate the purpose of the review should focus on the identification of unmet needs and to make onward referrals as indicated. The intention of identifying unmet needs is a key concept initially introduced into guidelines in response to a high proportion of stroke survivors reporting ongoing issues that were not managed via regular healthcare services [9]. More recent evidence demonstrates that the term *unmet need* covers a very large array of potential issues the stroke survivor (or their carers) may face, incorporating physical problems, mental health and well-being issues, impacted social participation, information requirements, and care needs [10]. Taking an all-encompassing approach to rectify these needs is perhaps overly ambitious to address in a single review appointment and attempts to do so risk creating an excessively cumbersome process. Consequently, the intended purpose of addressing unmet needs appears to have been interpreted differently in various service specifications [11–13] arguably contributing to the variation observed in service provision.

Previous studies have considered the intended purpose as part of wider evaluations of 6MR services. Abrahamson and Wilson [14] highlighted differences in how purpose was operationalised by different provider organisations, with specialist nurses providing a more medicalised review, and third sector organisations providing a more socially orientated review. They also reported that stroke survivors were often unclear about the purpose of the 6MR, a finding supported by a service evaluation in the South-West of England [15].

The lack of a clearly defined and consistently communicated purpose has likely led to variability in delivery, misaligned expectations, and suboptimal outcomes. To address this, it is essential to examine how different interest-holders define and operationalise the 6MR in practice. This understanding will strengthen programme theory by clarifying which outcomes should be prioritised and identifying the mechanisms through which the 6MR may achieve its intended effects.

The current study aimed to better understand the purpose of the 6MR by comparing and contrasting the perspectives of different interest-holder groups, and to identify what impact these perspectives have on the functioning of 6MR services.

## Methods

### Design

This study forms part of a larger qualitative study embedded within an explanatory-sequential mixed methods project (BE MoRe: Exploring the Benefits and Expectations of the 6-Month Review for Stroke Survivors). The initial quantitative phase of the project has been discussed elsewhere [6]. The qualitative phase of the project involved a multiple case study design utilising semi-structured interviews, clinical observations and documentary analysis, with an overarching aim of developing programme theory to understand how the 6MR operates. While the broader qualitative aspect of the project used a multiple case study design to capture contextual variation across three different sites, the present paper focuses on a sub-analysis of the qualitative data to explore interest-holders' perspectives on the purpose of the 6MR. Ethical approval for the study was obtained from the Health Research Authority (REC Reference: 24/WA/0059). This article has been structured in accordance with the Standards for Reporting Qualitative Research reporting guidelines [16].

Paradigmatically, this study takes a pragmatic stance, acknowledging that, for the current research questions, knowledge is constructed through interactions within the research process rather than being information waiting to be discovered. The pragmatic perspective emphasises generating knowledge that can be applied to real-world problems, highlighting the practical relevance of findings. As such, we hope to develop insights that are sensitive to context and can usefully inform practice and policy in a range of settings.

The lead researcher (RH) is a PhD candidate and a physiotherapist with over 15 years of experience working with stroke survivors in both acute and outpatient settings. He has also served as clinical lead for the commissioning of 6MR services, providing clinical expertise to commissioners during procurement. This background facilitated rapport-building during interviews and provided a nuanced understanding of system-level influences on service implementation. However, his clinical background may have influenced his perspective on non-clinically led services. To address this, he engaged in reflexivity throughout by maintaining a reflexive journal and participating in regular debriefing meetings with co-authors. The research team also consisted of two clinical academic physiotherapists (SA and LC), with expertise in stroke rehabilitation and implementation science research, and a social scientist (DG), with expertise in qualitative methods. None of the research team had prior professional relationships with staff at any of the study sites.

### Sample

Three 6MR services in England were included in the study. These services were purposively selected from the quantitative phase [6] of the project, in which services had left their contact details if interested in participating in the qualitative phase. Case selection aimed to maximise diversity across contextual factors such as the local geography, local demographic profile, type of provider organisation and service delivery model. Site A was a community NHS trust in the East

of England; Site B an acute NHS trust in the South East; and Site C a third sector organisation in the North West. Further detail of the sites can be seen in Supporting Information (S1 Table).

Participants were recruited from three interest-holder groups linked to three 6MR services in England: Service Users (including both stroke survivors and their carers); Service Providers (staff members actively delivering the 6MR); and Service Influencers (which included service managers, commissioners, and regional leaders in stroke care). At each site, the sample size was pragmatically estimated to include six to eight service users, and two to three from each of the service provider and service influencer groups. The final sample size was determined reflexively by the research team, taking into account the service structure of each site and continuing recruitment until it was felt that sufficient data had been gathered to capture the key nuances of each site.

Service users were included within the study if they were over 18 years and a stroke survivor, or the carer of a stroke survivor, who had had their 6MR in the last three months. Efforts were made to ensure stroke survivors with cognitive or communication difficulties could be included in the study by allowing extra time during the interview, using carers to support communication and memory, and seeking advice from a speech and language therapist when required. However, stroke survivors who were unable to provide consent or were unable to use a consistent method of communication (verbal or otherwise) were excluded from the study. Service users were provided with the study information sheet and invited to take part in the study by their 6MR provider. Interested participants either contacted the research team directly or agreed to be approached by the lead researcher. Service providers were purposively approached at the inception of each case. Service influencers were suggested by each service lead or purposively approached by the lead researcher depending on the nuances of each case. All consent procedures were undertaken by the lead researcher either face-to-face or virtually. Written consent was obtained for face-to-face interviews, while recorded verbal consent was used for virtual interviews where written consent was not feasible.

## Data collection

Semi-structured interviews were undertaken by the lead researcher either virtually or within the participant's home. All interviews were audio-recorded and transcribed verbatim. Interview schedules were developed for each of the interest-holder groups (S1 File). While each guide was tailored to the interest-holder group, interviews broadly explored perceptions of the purpose of the 6MR, the role of the 6MR within the wider stroke pathway, and views on outcomes such as benefit, value, and acceptability. Observations of the 6MR in action were undertaken at all three sites by the lead researcher. Observing the interactions between the service provider and service users enabled a richer understanding of the 6MR process by adding depth and context to reported experiences. Observations included all formats of service delivery such as telephone (with the provider and user communicating via speakerphone), virtual (with RH logged in to the call with the camera off) and face-to-face in either a clinic setting, the stroke survivor's own home or a nursing/residential home. All observations were undertaken with the researcher as a passive observer and details recorded contemporaneously using a structured observation proforma (S1 File). Documents related to the service were also analysed to allow a deeper understanding of the context shaping each service. Documentary data included service specifications, demographic information from freely accessible government databases, service evaluations and quality improvement reports, audit data, and service-related paperwork (i.e., referral forms, data collection tools).

Data were collected between June 2024 and April 2025. All anonymised interview transcripts, service-related documents, observation reports, and reflexive field notes were uploaded to NVivo 14 for analysis.

## Data analysis

A reflexive thematic analysis following Braun and Clarke's six-stage approach [17] was conducted, with each interest-holder group analysed separately. During analysis, a broad interpretation of purpose was adopted to capture

interest-holders' views on both the intended functions and the perceived benefits of the 6MR. Data from interviews, observations, and documents were included in the analysis. The lead researcher first familiarised themselves with the data by listening to recorded interviews and reading transcripts and service-related documents multiple times. Initial codes were generated using a line-by-line, inductive process. Initial codes were iteratively reviewed and collated into broader themes, which were refined through repeatedly returning to the data to ensure they were representative patterns of meaning. Consistent with reflexive thematic analysis, the lead researcher coded all transcripts independently. Regular reflexive meetings were held with the research team to critically question and refine coding decisions and theme generation [18].

Steps have been taken to ensure the anonymity of both the services and individuals involved. This included the use of pseudonyms, the omission of job titles in cases where individuals could be identified and limiting the detail of the location of services.

## Results

### Participants

A total sample of 36 participants took part in the study, comprising eight Service Providers, six Service Influencers and 22 Service Users (consisting of 19 stroke survivors and three family members). The composition of study participants at each site can be seen in Table 1. Data was collected from a total of 23 hours of interviews, 17 hours of observed practice and 23 documents related to the services.

The service provider group was comprised of four stroke specialist nurses, two therapists and two third sector employees. To protect the anonymity of the service influencers, it is not possible to disclose the specific roles they held. Members of this group often took on multiple roles; all were considered regional leaders within their respective locations. One member had direct involvement in the commissioning process, two had previous or current experience delivering 6MRs, and one was in the direct line of management at one of the sites. The demographic data for service users can be seen in Table 2. In instances where a family member was interviewed as a proxy, the demographic information presented in the table pertains to the stroke survivor.

### Findings

Seven themes were generated from the three interest-holder groups.

**Service providers.** All service providers highlighted that a main purpose of the 6MR was to identify unmet needs, echoing the intention in guidelines. However, at times, this intention was discordant with what they described and whilst 'identifying unmet needs' was the superficial headline they verbalised, thematic analysis revealed broader intentions in terms of purpose. Two themes were generated from the *Service Provider* group: *Mind the gap* and *Just passing through?* **Mind the gap:** Providers expressed that the 6MR should not be merely a process-driven, completion of a checklist. Instead, they reported a desire to deliver a holistic, person-centred service that was able to accommodate whatever needs the stroke survivor had. For some, this person-centred approach was prioritised above all else, with staff even showing a willingness to adapt rules if their patients benefitted.

**Table 1. Frequency of each interest-holder group at each site.**

|        | Service Providers | Service Influencers | Service Users |
|--------|-------------------|---------------------|---------------|
| Site A | 3                 | 2                   | 8             |
| Site B | 3                 | 3                   | 6             |
| Site C | 2                 | 1                   | 8             |
| *Total* | *8*              | *6*                 | *22*          |

**Table 2. Demographic data of service users.**

|  |  | *n* = 22 (3) |
|---|---|---|
| Sex | Male | 12 (1) |
|  | Female | 10 (2) |
| Age | <55 | 2 |
|  | 56–65 | 5 |
|  | 66–75 | 7 (1) |
|  | 76–85 | 6 (1) |
|  | >86 | 2 (1) |
| Care needs | Independent | 16 (2) |
|  | Informal care | 4 (1) |
|  | Formal care | 2 |
| Ethnicity | White (British) | 20 (2) |
|  | White (Other) | 2 (1) |
| Employment status | Employed | 3 |
|  | Unemployed | 2 |
|  | Sick leave | 3 |
|  | Retired | 14 (3) |

NB. Figures in brackets indicate how many of the total responses in each category were provided by family members interviewed as proxies. All demographic data refer to the stroke survivor, not the proxy.

> *…it's what's important to them, rather than me kind of going "oh you must fill this in".*

> *(Annabel, Stroke Specialist Nurse, Site A)*

> *…if, after that session, there are things that I need to do to follow up, we are not commissioned for it, but I doubt very much if any of us would not do it.*

> *(Grace, Stroke Specialist Nurse, Site A)*

All providers emphasised that the 6MR was more than just a chance to identify unmet needs. They felt it presented a real opportunity to share dedicated time with the stroke survivor, allowing them to tell their story. Providers assigned high value to this, reporting a cathartic experience as stroke survivors and carers felt heard, supported, and reassured.

> *…that's quite a powerful element that I've kind of thought about over the years really, it's the first time somebody's actually given them time and space to actually talk about what happened to them.*

> *(Annabel, Stroke Specialist Nurse, Site A)*

> *It's a bit fluffy, but, but I think it's important that people feel heard.*

> *(Grace, Stroke Specialist Nurse, Site A)*

Through this 'story-telling', providers described the 6MR as a moment whereby stroke survivors can gain direct benefit, rather than viewing the appointment as an administrative exercise in identifying needs for future interventions. They reported developing a deeper, human connection and partnership with the stroke survivor, working together to problem-solve. Some providers felt this element was lost when the review was not provided in person.

*You know, there's that personal element that, you know, we as humans enjoy, don't we? And it just makes it easier then to receive support.*

*(Ceri, Stroke Support Co-ordinator, Site C)*

However, at some sites, the desire to be person-centred could be constrained by their system's focus on efficiency and throughput. This created a clear gap between their intentions and what was practically achievable. At times, this resulted in a 'tick-box' exercise, focused more on meeting audit requirements than on meaningful engagement

*It probably does feel a little bit like a tick box in some circumstances, that we're just doing it because we're told that it should be done.*

*(Abby, Therapist, Site B)*

**Just passing through?:** Providers highlighted the dichotomy of purpose in relation to whether the 6MR was *actively* aiding recovery or *passively* gatekeeping to other services.

All providers saw the 6MR as an opportunity to provide relevant information or to sign-post the stroke survivor to services that met their needs. However, for some, this was seen as the limit of their input, viewing the 6MR as a passing checkpoint within the stroke pathway.

*…you're quite limited in what you can do. So, I don't know, I think sometimes the patients want you to actually do something for them, but you can't.*

*(Beth, Therapist, Site B)*

Some providers highlighted that, despite the 6MR being a one-off appointment, it was an opportunity to provide meaningful interactions. Commonly, this was described in terms of managing expectations and providing education in a safe and supportive manner. Providers identified the 6MR as an ideal time for this kind of intervention.

*…it's a good opportunity to have those conversations with patients, to be honest about recovery…*

*(Dana, Stroke Specialist Nurse, Site A)*

*…they can talk to someone outside the family, not emotionally involved, able to give information, advice, and support and challenge if appropriate.*

*(Scout, Stroke Support Co-ordinator, Site C)*

Occasionally, the 6MR was described as a negotiation. Providers discussed working with the stroke survivor and their loved ones to produce behaviour change when required. To achieve this, they reported needing to adapt their approach to the given situation, at different times being a directive instructor, a supportive coach or even a counsellor.

*We are, sometimes, marriage counsellors as well so that's always a (laughter)…… that happens fairly frequently.*

*(Annabel, Stroke Specialist Nurse, Site A)*

**Service influencers.**  The *Service Influencer* group emphasised a more expansive interpretation of purpose than *Service Providers*, more often considering the 6MR from a systems-level perspective. Two themes were generated from the data: *Charting the course* and *Getting things back on track*.

**Charting the course:** This group highlighted that a key purpose of the 6MR was to collect data on the population to inform or influence service improvement at a system- or regional-level. They felt that having a standardised point for this data collection presented an opportune moment to achieve an understanding of the needs of stroke survivors within the local population.

*It's almost like we have to try and compare apples with apples. And so I think it is, it's an attempt to, in a very complex, multifactorial delivery, complex intervention, it's, you know, a way of collecting data at a single time point that tries to give consensus and understanding of that fragmentation of what happens, but how much needs still exist for people.*

*(Lisa, Site A)*

*…what data do we have that we can use to compare and understand what's going on in our patch?*

*(Libby, Site B)*

Influencers discussed a desire to use this understanding to influence change within their networks. They felt that the data related to ongoing needs could be used as a powerful and convincing lever for change.

*There is also another level around, you know, can you make a… can you make a cost-effective argument by collecting your EQ-5D [EuroQol-5D - a standardised measure of health-related quality of life] that links into your QALYs [quality-adjusted life years – a measure in health economics] to then say to the system, this is what you've got to do then.*

*(Lisa, Site A)*

*…how that can then help guide us to what services are then... we need to commission in the future.*

*(Bridie, Site B)*

*…we look at what the data is telling us about, what the needs of stroke survivors are at that time, and try and use that to say, okay, well, there's clear patterns here, but for example, fatigue seems to be a real issue at six months. Can we look at what that's telling us and what services we therefore might need to provide and when?*

*(Cindy, Site C)*

This group were also cognisant of the impact that current services, or any future service changes, could have on other areas of the pathway. They saw the 6MR as having the potential to reduce burden elsewhere in the system and were critical of services that risked increasing burden.

*…I've always been interested in… to understand how many of those patients are referred back into primary care, potentially. Is that helping primary care to deliver that, or is it just overloading an already very busy service?*

*(Bridie, Site B)*

**Getting things back on track:** In contrast to some service providers, who reported a potential to actively treat stroke survivors during the review, service influencers saw the 6MR as an ideal time to assess and signpost onwards to other services. As such, they viewed the 6MR providers as having a gatekeeper role that filtered the unmet needs into other services.

*…it's not actually that the individual can solve or do all of those things […] it's about, again, coordinating services or signposting services that that person can access to hopefully meet the needs or those issues that are existing.*

*(Isobel, Site B)*

This ability to link in with other services was seen as a crucial role, and it was emphasised that the review process alone is insufficient if it does not lead to change for the patient.

*…measuring the cow doesn't make it any fatter! So, if we just review people, that doesn't help them. So, the review itself is… for me, the most important thing about it is that it enables us to continue to meet the changing needs of stroke survivors and their carers…*

*(Cindy, Site C)*

The service influencer group highlighted the potential that needs may have been missed within the stroke pathway, or needs may have evolved or surfaced since leaving coordinated services. They recognised a key function of the 6MR was to provide a safety net so that new or missed problems could be recognised and steps taken to address them. They specifically highlighted a need to ensure secondary prevention strategies were in place, again viewing this as something that would have implications on the wider system if not managed effectively.

*I do think the secondary prevention is an important area in relation to medication management, blood pressure and AF [atrial fibrillation] particularly.*

*(Cindy, Site C)*

*…so I would say the medication, taking the medication addressing the cardiovascular risks and talking them through how they can do that.*

*(Bridie, Site B)*

### Service users

Members of the service user group had varied and often opposing views of the purpose of the 6MR. This seemed to be in relation to how their individual experiences shaped their understanding retrospectively given that they had relatively narrow understanding of the concept of 6MRs prospectively. Three themes were generated from the service user data: *'Tickets please!', End of the line,* and *Safe to proceed*.

**'Tickets please!':** At face value, service users felt the 6MR was a routine, scheduled check-up to assess their progress and consider support for the future.

*I think it was a good way of measuring where I've been, where I was at, and deciding if I needed future help and pointing me in the direction of that.*

*(Phillip, Stroke Survivor, Site A)*

Prior to the 6MR, it was not always considered to be a particularly noteworthy appointment, and many service users were unfamiliar with the term 'six-month review'. They did not view it as a key part of the stroke pathway in the same grandeur as is emphasised in guidelines.

*I thought it was just to make sure that everything was all right, which I think was, you know, a nice, nice gesture.*

*(Jeremy, Stroke Survivor, Site A)*

Consequently, some stroke survivors were unprepared for the review and were not clear on its purpose. This element was improved when service users were provided with information pre-appointment, though this was inconsistent.

**End of the line:** Some service users saw the 6MR as the final point in their stroke pathway. The moment where their access to formalised support ended.

> *…basically, it was whatever the system has organised. It was the final thing, yes? Ticking off, finals, yes?*
>
> *(Ursula, Stroke Survivor, Site B)*

> *…that's the end of the line there, we've done that, you're done.*
>
> *(Ruby, Stroke Survivor, Site C)*

This feeling of reaching the end of the line came with both positive and negative connotations depending on the individual and their situation. From a negative perspective, some respondents highlighted a sense of superfluity, seeing no value in the review that they received.

> *So the six-months review was, in a way, superfluous, but nice to know that somebody still carried on, I guess.*
>
> *(Jeremy, Stroke Survivor, Site A)*

> *It's not like a major thing is it, in that it's going to make any changes, particularly – it's just touching base…*
>
> *(Ruby, Stroke Survivor, Site C)*

There was even a consideration that the 6MR was not for them as individuals but was instead to help the service provider gather feedback.

> *She really was auditing, if you like, what's going on in those six months…*
>
> *(Patrick, Husband of Stroke Survivor, Site C)*

However, others saw this end point in a more positive light. They retrospectively viewed it as a defining moment in their journey that marked the completion of one aspect of the pathway or as a transition point between formal healthcare services and reintegrating into society. Some service users felt the 6MR was part of a 'signing off' that then allowed them to move forward with the next stage of their lives.

> *But the six-month review and the stroke team before seemed to be at a point where, okay, it's over to you and your GP now to see how the future lies.*
>
> *(Phillip, Stroke Survivor, Site A)*

> *I'm responsible for my care. […] So I take on the responsibility of contacting someone if I need to.*
>
> *(Peter, Stroke Survivor, Site B)*

**Safe to proceed:** Service users saw the reassurance they gained from the 6MR as the key benefit. This seemed to be more important to them than other aspects, such as identifying their ongoing needs or making onwards referrals to other services. However, stroke survivors found this reassurance in different ways. Some stroke survivors felt reassurance merely from being in the 'system', feeling that someone was still looking out for them. This made them feel like they weren't alone in their recovery. They felt reassured by the time that they had been provided. In turn, this allowed them ample opportunity to go over their ongoing concerns and attain the answers they needed.

*Well, I think as well, just a bit of reassurance. I don't, I don't know kind of what I mean by that, um, but it was nice to have somebody to come and sit and have the time to sit and listen to the issues that we were having…*

*(Susan, Daughter of Stroke Survivor, Site C)*

Others found reassurance simply by having their efforts validated, with some using the 6MR as an opportunity to check that they were doing the right things.

*It was reassuring, I think. It's nice to have a third party say, oh, you're doing okay, or you need to concentrate on this or that. So yes, it was useful.*

*(Phillip, Stroke Survivor, Site A)*

A number of service users also had concerns about having another stroke. They expressed finding reassurance through discussing these concerns within the 6MR, even when the concerns weren't eradicated.

*I feel reassured. It was nice to be able to talk about all the things that concerned me. I wouldn't go to the GP with some of these things, you know?*

*(Janet, Stroke Survivor, Site A)*

Fig 1 presents the themes in an illustrative format and Table 3 provides a summary of the generated themes.

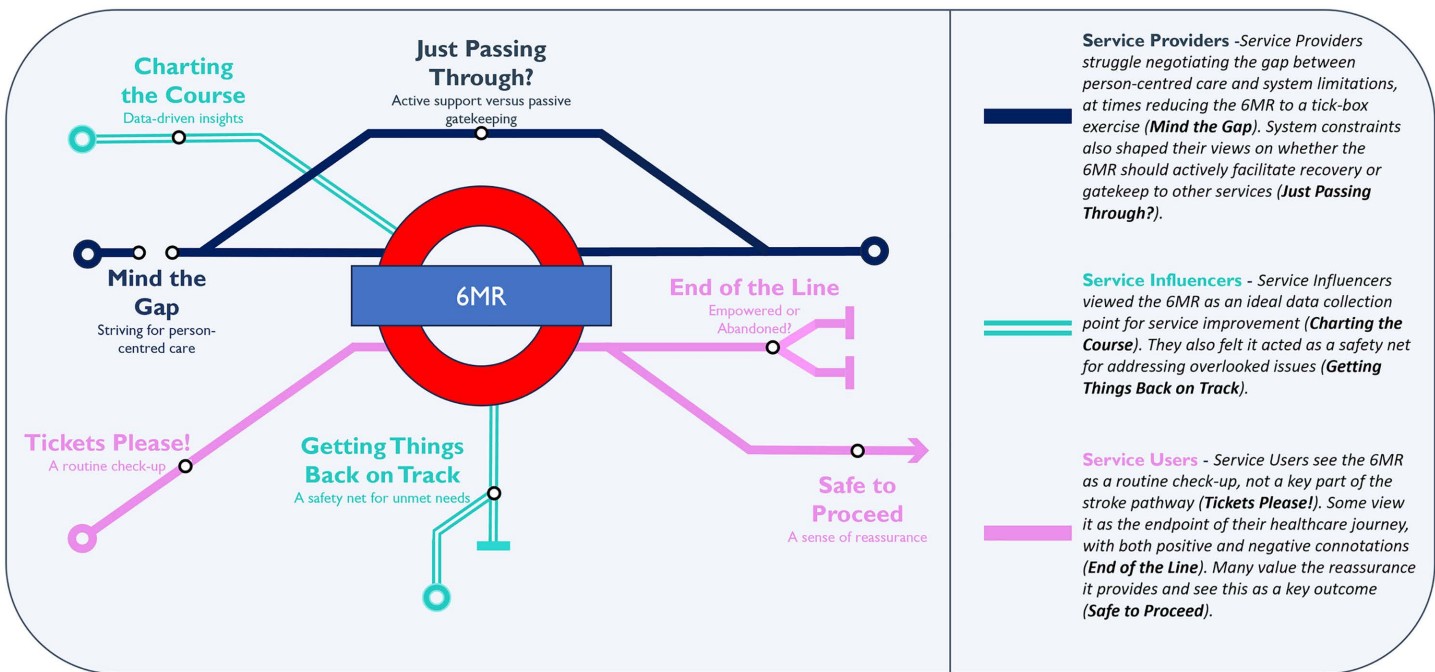

**Fig 1. Visual display of generated themes in relation to interest-holder groups.**

**Table 3. Summary table of generated themes.**

| | Themes | Overview of Theme | Example Quote |
|---|---|---|---|
| Service Providers | Mind the Gap | Providers strive to provide person-centred care but there is sometimes a gap between this intention and the limitations of the system. This gap can cause the purpose of the 6MR to denigrate to a tick-box exercise for audit requirements. | *It probably does feel a little bit like a tick box in some circumstances, that we're just doing it because we're told that it should be done.* (Abby, Therapist, Site B) |
| | Just Passing Through? | There was a dichotomy in views on whether the purpose of the 6MR was to channel stroke survivors into other services—acting as a gatekeeper—or to actively aid recovery by providing an intervention. Again, where providers positioned themselves along this continuum was shaped by the structural constraints and enablers within their respective systems. | *…you're quite limited in what you can do. So, I don't know, I think sometimes the patients want you to actually do something for them, but you can't.* (Beth, Therapist, Site B) |
| Service Influencers | Charting the Course: Data Driven Insights | Influencers believed a key purpose of the 6MR was to ensure effective, consistent data collection to inform future service improvement and to highlight service need. | *…we look at what the data is telling us about, what the needs of stroke survivors are at that time, and try and use that to say, okay, well, there's clear patterns here, but for example, fatigue seems to be a real issue at six months. Can we look at what that's telling us and what services we therefore might need to provide and when?* (Cindy, Site C) |
| | Getting Things Back on Track | Influencers felt the 6MR provided a 'safety net' that allowed ongoing or new issues to be highlighted. They saw this as an opportunity to get stroke survivors back into services to rectify any issues that may have been missed or neglected, with a view that this would cause less problems later. An important example of this was ensuring that secondary prevention measures are in place and effective. | *…it's not actually that the individual can solve or do all of those things […] it's about, again, coordinating services or signposting services that that person can access to hopefully meet the needs or those issues that are existing.* (Isobel, Site B) |
| Service Users | 'Tickets Please!': A Routine Check | Stroke survivors thought the 6MR was a routine check-up to see if they had any ongoing problems and to find solutions. They did not view it has a key aspect of the stroke pathway. | *I think it was a good way of measuring where I've been, where I was at, and deciding if I needed future help and pointing me in the direction of that.* (Phillip, Stroke Survivor, Site A) |
| | End of the Line | There was a view that the 6MR represented the end-point of the health-care input related to their stroke. Depending on the circumstances, and the individual, this end-point was seen for some as a positive that allowed them to move on with their lives, or as a negative that left them feeling frustrated or abandoned. | *…that's the end of the line there, we've done that, you're done.* (Ruby, Stroke Survivor, Site C) |
| | Safe to Proceed: A Sense of Reassurance | Service users' most emphasised and consistent understanding of purpose was that it provided individuals with a great deal of reassurance. They valued this aspect above all else. | *It was reassuring, I think. It's nice to have a third party say, oh, you're doing okay, or you need to concentrate on this or that. So yes, it was useful.* (Phillip, Stroke Survivor, Site A) |

## Discussion

Understanding *why* a service is provided is an essential first step in understanding how it should be implemented, measured, and evaluated, to ensure it is effectively achieving its intended outcomes. This study highlights that currently there is divergence of opinions between interest-holders, and no definitive agreed upon understanding of purpose.

This finding of variation in interest-holder perspectives adds to previous work exploring the purpose of the 6MR [14,15] by exploring how these perspectives shape the function of the 6MR and the experience for stroke survivors. Service providers strived to provide holistic, person-centred care but often found themselves limited by the demands of the system. Whereas, the service influencer group, made up of higher-level managers, commissioners, and regional leaders, took a view that the 6MR had a greater purpose in terms of reducing burden in the system and using the resultant data to guide future service initiatives. Service users often assumed the 6MR was a routine check-up and assigned little importance to it, but they valued the reassurance they gained from the experience. For this group, it was clear that their experience of the 6MR was shaped by how the context of the wider system interacted with their individual situation. Stroke survivors

found the 6MR most beneficial when they understood its purpose, had a provider who was knowledgeable, caring, and considered the 6MR as integrated within the wider system. Conversely, stroke survivors expressed frustration when the 6MR was unexpected and lacked clarity, and they perceived a lack of willingness or ability from providers to resolve their needs or concerns.

There is wide variation in the delivery of 6MRs within the UK [6]. It is suggested that how purpose is understood and operationalised by providers may, in part, explain this variation. Giddens' structuration theory [19] conveys the concept of the *duality of structure*, whereby the structure of a social phenomenon is not independent of human agency acting within it. Instead, there is a complex interaction in which structure (comprising of social rules, norms and values) both shapes, and is shaped by, human agency (through the interpretation and decision-making of individuals). In terms of the 6MR, the providers' interpretation of purpose may be enabled or constrained by the practices and shared ethos of the system in which they operate, and in turn their actions may influence the function of the system. This interaction informs how the 6MR is operationalised by providers in practice (Fig 2). Through feedback loops, how the 6MR is delivered may reinforce or alter the provider's understanding of its purpose. Considering the 6MR in this way may help to explain the variation seen in practice. With little clarity on the purpose of the 6MR, providers are required to generate their own interpretation based on their values and experience while being constrained by the context of the system in which they work.

Fig 2 illustrates Giddens' structuration theory as applied to the six-month review (6MR), showing the dynamic relationship between provider agency (interpretation and decision-making) and the structural context (shared rules, norms, and values). Feedback loops demonstrate how 6MR delivery may, in turn, influence providers' evolving understanding of its purpose.

Readers may have noted the array of train analogies in the themes. The naming of these themes arose inductively from the data rather than being preconceived; however, they do provide a fitting metaphor for the 6MR. Stroke survivors in the study described being on a journey: some had a clear plan of where they were heading, while others felt lost and frustrated, unsure where to go next. The 6MR was akin to an interchange station whereby some would be redirected in a new direction, others would continue on the same trajectory, and some would have reached their endpoint. At times, this left the individual reassured and satisfied they were heading in the right direction; at other times, they were left unfulfilled.

The sense of reassurance was the key benefit that stroke survivors described. Interestingly, this feeling did not seem to correlate with how many unmet needs were resolved or how many onward referrals were made. Instead, it was related to how they experienced the 6MR and how they were able to interact with the 6MR provider, a similar finding to other areas of healthcare [20–22]. The reassurance that stroke survivors experience may seem a trivial outcome, especially when considered against the cost of healthcare services. However, effective reassurance has been shown to improve patient empowerment [23], reduce symptom burden [22], and reduce the need for further healthcare utilisation [24]. As such,

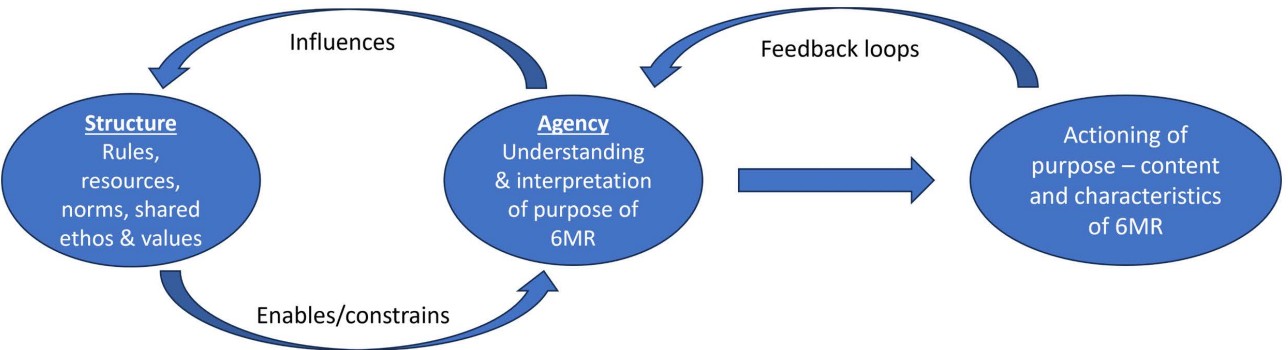

**Fig 2. Application of Giddens' *duality of structure* to the 6MR.**

patient activation and self-management may be increased. This is thought to be more apparent when stroke services are actively engaging patients through collaborative decision-making and problem solving, rather than passive services that lack continuity of care and provide limited information [25]. Therefore, achieving reassurance via a positive experience within the 6MR may be the mechanism that supports these outcomes. Whilst reassurance is not a specified intention of guidelines or service providers, nor an anticipated expectation of service users, it is an important aspect that warrants further exploration, especially from a health economics perspective. The role of reassurance may be more important than other intentions of the 6MR and may add more value than just the "fluffy stuff" that one participant reported.

## Strengths and limitations

A key strength of this study is the inclusion of stroke survivors with aphasia ($n = 4$) and cognitive impairments ($n = 2$); two groups often excluded from research. Their inclusion is an essential element of this research given that their experiences are likely to be shaped by additional contextual barriers.

However, the study is limited by the lack of ethnic diversity of participants recruited to the service user group, despite attempts to purposively sample at each site. This was in part due to the three selected sites serving populations whose proportions of White British individuals were much higher than national averages. The views of stroke survivors from other ethnicities would have been a valuable insight into the current study. White British patients are more likely to uptake NHS health checks [26] and, anecdotally, service providers perceive an inequity in access to the 6MR for ethnicities other than white [6]. Reasons for this are unclear but could be influenced by language and cultural barriers [27]. Consequently, the lack of ability within our methods to recruit stroke survivors who did not speak English was a further limitation. As was the inability to gather the views of stroke survivors who chose to not take up the offer of the 6MR. Their views on the perceived purpose of the 6MR may have provided a degree of explanation into their decision to refuse the 6MR. Further research to address these limitations is required to enrich understanding related to the inequitable access to the 6MR.

The study was also limited by the lack of service user participants who had high care needs. Instead, views were skewed more to those individuals who were managing the symptoms of their stroke independently. Data from stroke survivors with a higher level of need (such as those in nursing homes) would have added further depth to the analysis and there is a risk that this group may experience the 6MR in a way that was not captured in the current study. The use of multiple data sources helped to mitigate this to a degree by including this group in observational analysis.

## Conclusion

Interest-holder groups have different views on the purpose of the 6MR. Differences between service influencers' views (with a system-level focus) and service providers' views (with a patient-centred focus) have the potential to cause conflict through misaligned expectations. For example, commissioners prioritising data collection may lead to the review becoming procedural rather than holistic. Equally, differences in the ideation of the purpose of the 6MR between service providers and service users impacts the stroke survivor's experience of the 6MR, and, in turn, impacts the realisation of positive outcomes. Furthermore, this experience is influenced by how context shapes the operationalisation of the 6MR.

The function of the 6MR, and the underlying assumptions about how it delivers benefits for stroke survivors, is more complex than is reflected in current clinical guidelines. Clarifying its intended purpose for key interest-holders and clearly articulating the key elements and mechanisms through which it is expected to achieve outcomes can support the development of a robust programme theory. This is needed to facilitate more effective implementation, help align interest-holders' perspectives and reduce unwarranted variation in practice by providing commissioners and service providers with a clearer direction and shared understanding of the 6MR's goals. Our future work will focus on identifying the key ingredients of the 6MR and will further explore how outcomes emerge from the interaction of the intervention and the context.

## Supporting information

**S1 Table. Comparison table of the three sites.**
(DOCX)

**S1 File. Interview schedules for the three interest-holder groups and observation proforma.**
(DOCX)

**S2 File. Anonymised participant quotes and observational reflections by theme.**
(DOCX)

## Author contributions

**Conceptualization:** Rich Holmes, Suzanne Ackerley, Dawn Goodwin, Louise A Connell.

**Data curation:** Rich Holmes.

**Formal analysis:** Rich Holmes, Suzanne Ackerley, Dawn Goodwin, Louise A Connell.

**Funding acquisition:** Rich Holmes.

**Investigation:** Rich Holmes.

**Methodology:** Rich Holmes, Suzanne Ackerley, Dawn Goodwin, Louise A Connell.

**Project administration:** Rich Holmes.

**Resources:** Rich Holmes.

**Supervision:** Suzanne Ackerley, Dawn Goodwin, Louise A Connell.

**Visualization:** Rich Holmes.

**Writing – original draft:** Rich Holmes.

**Writing – review & editing:** Rich Holmes, Suzanne Ackerley, Dawn Goodwin, Louise A Connell.

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
