## [Decision Letter · Decision Letter 0]

30 Oct 2025

Dear Dr. Holmes,

Thank you for submitting your manuscript to PLOS ONE. After careful consideration, we feel that it has merit but does not fully meet PLOS ONE’s publication criteria as it currently stands. Therefore, we invite you to submit a revised version of the manuscript that addresses the points raised during the review process.

We look forward to receiving your revised manuscript.

Kind regards,

Sebastian Suarez Fuller, PhD

Academic Editor

PLOS ONE

Journal Requirements:

Reviewers' comments:

Reviewer's Responses to Questions

**Comments to the Author**

1. Is the manuscript technically sound, and do the data support the conclusions?

Reviewer #1: Yes

Reviewer #2: Partly

2. Has the statistical analysis been performed appropriately and rigorously?

Reviewer #1: N/A

Reviewer #2: N/A

3. Have the authors made all data underlying the findings in their manuscript fully available?

Reviewer #1: Yes

Reviewer #2: Yes

4. Is the manuscript presented in an intelligible fashion and written in standard English?

Reviewer #1: Yes

Reviewer #2: Yes

Reviewer #1: Thank you for the opportunity to review this clearly written and interesting article. I hope the comments below are of use in further improving this manuscript.

Abstract: It’s unclear from the abstract methods section that this submission only reports on one element of data collection i.e. the interviews. I would suggest removing reference to the other methods of data collection within the abstract.

You have used the term stakeholders within this submission. You may wish to use different terminology for a global audience such as interest holders. Please refer to the following reference for more information: Akl, EA, Khabsa, J, Petkovic, J, Magwood, O, Lytvyn, L, Motilall, A, Campbell, P, Todhunter-Brown, A, Schünemann, HJ, Welch, V, Tugwell, P & Concannon, TW 2024, '“Interest-holders”: A new term to replace “stakeholders” in the context of health research and policy', Cochrane Evidence Synthesis and Methods, vol. 2, no. 11, e70007. https://doi.org/10.1002/cesm.70007

Methods: Given the role of the lead researcher in 6MR development – does this mean you knew some of the staff prior to the interviews? Information on any relationships with participants would be helpful here as per the COREQ checklist.

More detail on how the reflexive thematic analysis was undertaken would be helpful e.g. did you undertake line by line coding to generate initial themes which were then refined?

It’s unclear what a collaborative reflexive approach was- can you please clarify?

It would be helpful to know how much time was spent on 6MR appointments and did this vary throughout different services to provide further context.

P15 abbreviations are used within the quote – it would be helpful to explain what these are for readers.

Discussion: It is noted that all your participants were white, I am pleased to see that this is acknowledged within your limitations but given that other ethnicities can have a higher prevalence of stroke and are less likely to attend 6MR perhaps a recommendation that further work should address this is needed?

Thank you for providing a summary table – this is helpful for readers. I also enjoyed your use of themes aligned to train travel. Finally, well done for including people with aphasia and cognitive deficits within this research.

Reviewer #2: ONE-D-25-27773 Clear tracks and missed connections: A qualitative study exploring how stakeholder perceptions of purpose shape the implementation and experience of the six-month review for stroke survivors. S

A pleasure to read but it will be even better with a little more work, and demonstrate clearly how the conclusions were reached. The main weakness is stating that it was a case study design but not demonstrating this in the results or discussion (unless I am misunderstanding your use of the term).

Introduction: covers the main points but might be worth mentioning that 6MR is part of process than includes review at 6weeks, 6 months and yearly, but that timeframe of reviews was arbitrary. You could then pick this up in the discussion.

Line 79-80 mentions unmet needs (McKevitt et al 2011) as rationale to support the 6MR but it would be worth adding a sentence to qualify if this is still the case (or more likely, that need is greater in current economic/healthcare climate) and any recent papers to support this.

Method:

Study Design, p5-6: states case study design (please clarify what type, there are several), tools (interview/obs/documentary analysis) and paradigm (pragmatic) but the rationale for all these, and how they worked together, needs stronger/clearer justification. Lines 117-9 are non-sequiturs.

While it states that it is a multiple case study design, this is not clear in the text – in line 132 it states ‘3 stakeholder groups linked to three 6MR services’ but this is not described (who, how were the sites recruited, why defining features).

Please add a description of the case study sites – this need to be added to include how sites were selected (and it’s ok to say it was whoever you could get), how they differ, the wider context, and a description of the model of 6MR for each site (e.g. who carries it out, where, how, whether they use a standardised tool or their own form etc). This information could be tabulated to save words. I’m assuming (from lines 120-125) that the author works in one of the sites but this isn’t clear.

What was the unit of analysis for the case study site (if you’re using Yin)?

Sample size: How many interviews did you intend to carry out for each group, and at each site?

What was your approach to data saturation – it’s ok to say you did what you could in the time you had but be explicit.

Recruitment: it’s not clear how services were approached and then how service users were recruited – lines 143-4 only state the end point, that staff/research invited them, but this misses how you got to that point.

Observations: unclear what/who/how you observed each of the formats (line 164), including phone calls. Nursing/residential homes are mentioned (p8, line 165) but not mentioned elsewhere – including data specific to NH/RHs would enhance the results.

Topic guides: worth including a summary of key topics in the text, and the full versions keep as supplementary.

Data analysis, p8: please add further information on:

- How you analysed observational data (within and across sites)

- How you analysed documents (within and across sites)

- How you carried out thematic analysis –more detail e.g. the nuances of how you compared across stakeholder groups – and between sites, which is absent.

- How you synthesised the data within & across the sites - obs, documents, interviews (staff, influences & patients/carers)

- How you used CICI framework

With further information on the above, it will be clearer how you have addressed rigour/trustworthiness.

Results

As above, it states that it is a multiple case study design but the analysis does not reflect this which makes it hard to comment on the results. While quotes are attributed to each site it is not clear how the sites differed and what these differences meant e.g. in terms of patient experience or outcomes. For example:

- p11 mentions that the 6MR shouldn’t be a checklist approach but it’s unclear how the reviews were carried out and what, if any (checklist style) tools were used.

- Bottom of p11 mentions the importance of dedicated time with the stroke survivor, but the reader does not know if the person carrying out the 6MR has met them before during the rehab phase, or carried out a 6 week review, or is going in cold. Nor do we know if patients/carers were able to contact the Reviewer afterwards with any queries – some clinicians provide a phone number, others don’t, but this is relevant & would come under descriptions of the 3 sites.

Once you have addressed the points under data analysis, then briefly present the key findings – you could cut by observations, documentary analysis, and interviews and/or by site but for case study design you need to be clear about differences/similarities between sites and this is lacking. You can integrate documentary & obs with interview data but be explicit what is from which category. You could even include notes from your field log.

Some of the sub-headings sound good but lack clarity/meaning – it might be worth expanding the descriptor e.g. ‘charting the course of……’. Under this theme it is unclear what data was collected, which would be useful to state under the theme or under the site descriptions.

I’m not convinced that Table 3 is useful – it’s not clear the interconnections between themes, as Fig 1 aims to represent. See when you’ve reviewed the results – the Table might be better as supplementary information.

Discussion

Overall, good points but needs to be further developed/refined, and ensure that all statement are backed up by what’s been presented in the results (see below).

Line 447 – mentions wide variation in delivery of 6MR in UK – as above, this needs to be mentioned in the introduction and lead into your choice of case study sites / types of 6MR.

Figure 1 makes good sense but Figure 2 needs to be further developed and explained/justified in the text. Currently, it doesn’t add to the text. In the same paragraph (lines 456-459), its proposed that systems with high value on the 6MR take one approach versus those that see it as low priority and take a different direction – but we can’t see that in the results, and you need to back it up – hence why it’s important to describe the three settings and the differences between them.

Study limitations – well covered.

Conclusion – this brings in programme theory which hasn’t been mentioned before and seems somewhat out of place (given it’s not a realist evaluation). The final sentence about future research would benefit from refining.

**Do you want your identity to be public for this peer review?** For information about this choice, including consent withdrawal, please see our Privacy Policy

Reviewer #1: No

Reviewer #2: **Yes: ** Dr Vanessa Abrahamson

---

## [Author Response · Author response to Decision Letter 1]

5 Nov 2025

Thank for your time and expertise in reviewing our manuscript.

Please see the attached document for a point-by-point response to your comments.

---

## [Editor Report · Decision Letter 1]

1 Dec 2025

Clear tracks or missed connections? A qualitative study exploring how interest-holder perceptions of purpose shape the implementation and experience of the six-month review for stroke survivors

PONE-D-25-27773R1

Dear Dr. Holmes,

We’re pleased to inform you that your manuscript has been judged scientifically suitable for publication and will be formally accepted for publication once it meets all outstanding technical requirements.

Kind regards,

Sebastian Suarez Fuller, PhD

Academic Editor

PLOS ONE

Additional Editor Comments (optional):

Thank you for your thoughtful attention to reviewer comments, and your patience with the publication process.
---

## [Editor Report · Acceptance letter]

PONE-D-25-27773R1

PLOS One

Dear Dr. Holmes,

I'm pleased to inform you that your manuscript has been deemed suitable for publication in PLOS One. Congratulations! Your manuscript is now being handed over to our production team.

Kind regards,

on behalf of

Dr. Sebastian Suarez Fuller

Academic Editor

PLOS One